# Features of Lipid Metabolism Disorders in Primary Biliary Cholangitis

**DOI:** 10.3390/biomedicines10123046

**Published:** 2022-11-25

**Authors:** Vasiliy I. Reshetnyak, Igor V. Maev

**Affiliations:** Department of Propaedeutics of Internal Diseases and Gastroenterology, A.I. Yevdokimov Moscow State University of Medicine and Dentistry, Moscow 127473, Russia

**Keywords:** primary biliary cholangitis (PBC), dietary lipid metabolism disorders in PBC, hypercholesterolemia, mechanism of dyslipidemia in PBC

## Abstract

Primary biliary cholangitis (PBC), previously known as primary biliary cirrhosis, is an autoimmune liver disease that mostly affects women. A progressive disorder in the processes of bile secretion and enterohepatic bile salts circulation in patients with PBC already in its early stages, leading to an insufficient release of bile acids into the bowel and their entry into the systemic circulation. Insufficient bile acids released into the duodenum contributes to the development of malabsorption, energy malnutrition, and slowly progressive weight loss. The pathophysiological mechanisms of weight loss and its slow progression are associated with the deterioration of the fat emulsification processes and with the reduced absorption of hydrolyzed products, such as fatty acids and monoglycerides, with steatorrhea in patients with PBC, as well as in those with gut dysbiosis. Just in the early stages of the disease, this results in accelerated fatty acid β-oxidation that is aimed at compensating for progressive energy malnutrition. The entry of bile acids into the systemic circulation in PBC is accompanied by dyslipidemia. The mechanism of hyperlipidemia in patients with PBC differs from that in other conditions because along with an increase in total cholesterol (TC), there are elevated high-density lipoprotein levels and the appearance of unusual lipoprotein X (Lp-X). The appearance of Lp-X is most likely to be the body’s protective reaction to inactivate the detergent effect of bile acids on the membrane structures of blood corpuscles and vascular endothelial cells. It is bile acids, rather than TC levels, that correlate with the content of Lp-X and determine its formation. Concomitant hypercholesterolemia in patients with PBC is also aimed at neutralizing the detergent effect of bile acids that have entered the systemic circulation and is most likely a compensatory reaction of the body. “Anomalous” hypercholesterolemia in PBC can serve as a model system for the search and development of new methods for the treatment of dyslipidemia since it occurs without an increase in the incidence of cardiovascular events.

## 1. Introduction

Primary biliary cholangitis (PBC), previously known as primary biliary cirrhosis, is a chronic progressive cholestatic granulomatous and a destructive inflammatory lesion of small intralobular and septal bile ducts, which is likely to be caused by an autoimmune mechanism in the presence of serum antimitochondrial antibodies and a potential tendency to progress to cirrhosis [1,2,3]. Autoimmune tolerance defects are critical factors in disease initiation and the gradual development of intrahepatic cholestasis [3,4]. The latter in PBC is associated with damage to subcellular structures in the intrahepatic bile duct epithelial cells and with a change in the metabolism of bile acids due to the disrupted processes of bile secretion and their enterohepatic circulation. Progressive intrahepatic cholestasis results in an insufficient release of bile acids into the duodenum and, on the other hand, their increased accumulation in the hepatocytes and plasma. It is precisely these bile production and excretion changes that should be considered as an underlying cause of lipid metabolism disorders in PBC. In this case, lipid metabolism and transport disorders occur with dietary lipids as well as lipids and their transporting systems synthesized in the body.

## 2. The Mechanism of Dietary Lipid Metabolism Disorders in PBC

Insufficient release of bile acids into the duodenum in patients with PBC leads to a decrease in the rate of hydrolysis of fats and a reduction in the absorption of fats and fat-soluble vitamins. This contributes to the development of steatorrhea, protein–energy, and vitamin–mineral malnutrition. [5,6,7,8]. The seriousness of steatorrhea correlates with the decreased excretion and concentration of bile acids in the intestinal lumen (r = 0.82; *p* < 0.0001), with the level of a serum bilirubin increase (r = 0.88; *p* < 0.001), and with the late histological stages of PBC (*p* < 0.005) [9]. Insufficient emulsification of fats underlies the mechanism of the development of steatorrhea [6]. At the same time, the exocrine function of the pancreas is not disturbed. The lipases synthesized by the pancreas are involved in the hydrolysis of insufficiently emulsified fats, which slows down the rate of their splitting. The results obtained by Ross et al. indicate that pancreatic synthesis of lipases and their entry into the duodenum in PBC is not impaired and is not the cause of steatorrhea development [10]. The activity of pancreatic amylase in patients with PBC is within the normal range [10,11]. The severity of steatorrhea also does not correlate with the increased activity of alkaline phosphatase in these patients [10,11]. 

In the intestine, bile acids are taken part in the absorption of fat-soluble vitamins and fat hydrolysis products. Monoglycerides and fatty acids formed as a result of the hydrolysis of triglycerides and phospholipids, as well as fat-soluble vitamins together with bile acids, form lipoid-bile complexes that are absorbed by enterocytes (Figure 1). The latter are formed due to the detergent properties of bile acids, which are able to solubilize fatty acids and monoglycerides with the formation of micellar or lamellar structures [5,7,8]. Inside the enterocytes, the complexes break down. Fatty acids are used by the enterocytes as a building and/or energy material, or together with tri-, di-, and mono-glycerides and phospholipids form chylomicrons for further transport through the body. At the same time, bile acids again can be released into the intestinal lumen and take part in the processes of emulsification and absorption of fats and fat-soluble vitamins. This process can be repeated 4–6 times during the passage of bile acids through the intestine [12]. 

The insufficient release of primary bile acids in the intestine in PBС leads to a change in the composition of the microbiome, causing gut dysbiosis [10,13,14,15]. According to DiBaise et al., dysbiosis in patients with PBC also determines the development of steatorrhea [16]. Therefore, these patients should be necessarily evaluated for bacterial overgrowth [16].

Since insufficient bile acid release into the bowel is one of the first signs, impurities of partially digested fecal fats can be detected in patients with PBC already at an early stage. Steatorrhea in PBC is manifested with diarrhea varying in severity. At the same time, some patients with PBC may experience constipation. The development of the latter may be associated with a deficiency of bile acids that stimulate intestinal motility and gut dysbiosis [13,17,18].

Gradually and imperceptibly progressing steatorrhea leads to malnutrition and slowly progressive weight loss in patients with PBC [8], which is manifested only by general weakness and/or lower performance for quite a long time [19,20,21]. Even a slight nutrient deficiency is accompanied by gradually progressing glycogenolysis and a reduction in glycogenogenesis, which results in the activation of compensatory mechanisms. The latter is aimed at protecting vital organs that need higher energy consumption [22]. As a result, reserves of adipose tissue are used as an energy material. The use of fatty acids as an energy material and activation of β-oxidation of fatty acids is accompanied by the development of slowly progressive weight loss and malnutrition in patients with PBC [8,23]. 

## 3. Hypercholesterolemia and Xanthelasmas in PBC

The poor release of bile acids into the intestine in PBC impairs dietary cholesterol solubilization and micelle formation, which reduces intestinal cholesterol absorption. Through the feedback system, intrahepatic cholesterol synthesis is stimulated and the uptake of low-density lipoproteins (LDLs) by the liver is reduced through their receptors [24]. Increased hepatic synthesis of cholesterol causes its blood level to be elevated.

Due to developing and progressing cholestasis, there is a simultaneous hepatocyte accumulation of bile acids and an increase in their plasma level. Bile acids that have entered the systemic circulation in patients with PBC call for neutralization of their detergent effect on the membrane structures of blood corpuscles and vascular wall endothelial cells. This is accompanied by impaired lipid metabolism in patients with PBC. The resulting dyslipidemia is associated primarily with changes in the synthesis and transport of cholesterol and phospholipids (PhLs). Increasing plasma bile acid levels through the feedback system in patients with PBC gives rise to the higher expression of fibroblast growth factor 19 (FGF19) that activates the hepatic membrane receptor FGFR4 by suppressing the synthesis of bile acids, which contributes to the development of hypercholesterolemia [25,26,27,28]. 

In patients with PBC, long-term elevated plasma levels of cholesterol as a result of its hepatic synthesis can lead to xanthomas and xanthelasmas. Xanthelasmas exhibit a variety of shapes, single or multiple, flat, and light yellow bumps that are slightly raised above the skin.

Xanthelasmas appear in patients with PBC on the skin of the upper and lower eyelids, in the palmar creases, and under the breast. Along with this, xanthomas can be detected in the area of joints, tendons, and at the points exposed to frequent prolonged pressure (in the area of the elbow and knee joints, and buttocks). 

There is a relationship between the development of cutaneous xanthelasmas and the elevated serum level of total cholesterol. According to Kunkel HG and Ahrens EH, xanthelasmas on the skin appear when the blood cholesterol concentration is more than 450 mg/dL [29]. At the same time, the elevated plasma cholesterol level should persist for at least 3 months. Xanthelasmas may disappear when the cholesterol levels become normal, as well as in the late stage of the disease due to the progression of hepatocellular damage and impaired cholesterol synthesis. Low-fat diets to reduce xanthelasmas have been found to be unsuccessful and even harmful [30].

Biochemical tests of early-stage PBC patients’ sera revealed dyslipidemia manifested by elevated levels of TC, PhLs, fatty acids, LDL cholesterol (LDL-C), and high-density lipoprotein (HDL) cholesterol (HDL-C) [31], as well as by the appearance of plasma lipoprotein X (Lp-X) [25]. Moreover, the level of triglycerides in patients with PBC remains practically unchanged or slightly increased [32]. It is assumed that a slight increase in neutral lipids in PBC may be associated with the decreased activity of lipoprotein (LP) lipase [32]. The latter splits triglycerides contained in the largest and lipid-rich LPs (chylomicrons (CMs)) and very low-density lipoproteins (VLDL).

## 4. The Mechanism of Dyslipidemia in PBC

To understand the mechanisms of lipid metabolism disorders in PBC, it is important to discuss the physicochemical properties of major lipids and their transport in the human body. 

It has been known that triglycerides, PhLs, cholesterol, and fatty acids in an aqueous medium, such as blood and bile, cannot exist in monomeric form. Therefore, lipids are transported in the blood and bile through the formation of special micellar and lamellar structures, LPs. Since among all the above lipids, triglyceride and cholesterol molecules are the most hydrophobic, they cannot form micellar and lamellar structures. Whereas, being amphiphilic compounds, PhLs (containing hydrophilic and hydrophobic parts in their molecule), readily form micellar and/or lamellar (monolayer or bilayer) structures, such as different classes of blood LPs. Blood LPs are known to be unilamellar phospholipid particles [33,34,35]. The latter can solubilize (capture) non-esterified (free) cholesterol (a large steroid structure with a small charged hydroxyl group) unto themselves and integrate it into the phospholipid particle between phospholipid molecules. In this regard, PhLs in the LP particles can be deemed as a solvent of free cholesterol (Figure 2). The cholesterol/PhL ratio in the plasma LPs along with the molecular weight of LPs (HDL, LDL, or VLDL) predetermines the degree of cholesterol solubility. Esterified (fatty acid-bound) cholesterol and triglycerides that are completely hydrophobic molecules are located inside (in the core) of a PhL monolayer particle, and thus are transported in plasma.

In addition to cholesterol and PhLs (mainly phosphatidylcholines and lecithins), bile acids, the end product of cholesterol catabolism, are present in bile. Bile acids, such as PhLs, are amphiphilic compounds and can also form micellar and lamellar structures in an aqueous medium and serve as a solvent for cholesterol. As a consequence, PhLs and bile acids in bile form micellar and lamellar structures that solubilize cholesterol onto themselves (Figure 2). The degree of cholesterol solubility in bile is determined by the ratio of cholesterol/PhLs/bile acids. In addition to PhLs, bile acids present in these particles give rise to PhL bilayer/multilayer particles [36,37] that in native bile represent bile LPs [37]. The total concentration of bile acids in the isolated bile LPs ranges from 1 to 3% in weight [37]. Albumin in bile LP functions as an apoprotein [37]. Thus, bile LPs differ from plasma LPs in their structural organization. The former are PhL bilayer/multilayer particles and the latter are monolayer ones, which is determined fundamentally by the presence of bile acids in bile and by their absence in plasma, as well as by the low albumin concentration in bile [36,37]. Biliary excretion of cholesterol and bile acids plays an important role in cholesterol homeostasis in humans. Normally, both bile LP and plasma ones in healthy individuals do not intersect and perform an important function, such as the transport of lipids and main cholesterol in plasma and bile (Figure 2). It is important to emphasize that in this case, PhLs serve as a solvent of cholesterol in blood LP, while PhLs and bile acids do this in bile, which determines the monolayer and bilayer/multilayer structure of the former and the latter, respectively.

The homeostasis of cholesterol involves its movement between peripheral tissues and the liver. Maintaining a constant plasma level of cholesterol in a healthy individual depends on its intake, transport, and excretion from the systemic circulation. The content of plasma cholesterol is replenished by the intestinal absorption of dietary and biliary cholesterol, by the synthesis of endogenous cholesterol that is synthesized predominantly in the liver, and by the secretion of cholesterol-containing VLDL and LDL particles into the bloodstream [35]. The reduced plasma cholesterol level is achieved by the hepatic uptake of LDL-C and HDL-C, by the excretion of biliary cholesterol and its catabolism in the liver to form primary (cholic and chenodeoxycholic) bile acids [38]. A small amount of cholesterol is lost along with skin and intestinal epithelial cells that naturally exfoliate from the surface of the skin and mucous membranes [39]. The liver is the central organ that regulates the de novo biosynthesis of cholesterol, its excretion into bile (directly or after conversion to bile acids), its secretion into the blood as VLDL and LDL, the modulation of receptor-mediated cellular uptake of cholesterol, the formation of cholesterol esters that are more hydrophobic than cholesterol itself, and the storage of cholesterol.

Quantitative changes in bile acids as one of the solvents of intestinal and plasma cholesterol in PBC patients alter the conditions for the transport of cholesterol and its excretion from the body.

Based on the physicochemical properties of bile acids, it is possible to neutralize the latter that have entered the systemic circulation as a result of cholestasis in PBC patients, giving rise to micellar and lamellar structures with PhLs and cholesterol, similar to LPs, which form in bile. As a result, in the plasma of patients with PBC already in its early stages, the levels of PhLs, TC, VLDL-C, LDL-C, and HDL-C increase, and an abnormal Lp-X is detected [40]. The elevated HDL-C levels in PBC patients depend in part upon apolipoprotein A1 (ApoA-1) [41,42]. ApoA-1, the major protein component of HDL, incorporates two main classes of LPs that contain ApoA-1: LpA1 and LpA1:A2 [43]. LpA1 is found in greater concentrations than LpA1:A2 in PBC patients versus the control group [43]. Reduced HDL-C concentrations due to a significant decrease in the protein synthetic function of hepatocytes may be observed in late (terminal)-stage PBC patients having Lp-X [40]. 

Lp-X is considered to be an abnormal LDL that is present in patients with intrahepatic or extrahepatic cholestasis [44,45,46,47]. Lp-X contains bile acids, albumin, a high proportion of non-esterified (free) cholesterol, and PhLs [36]. Unfortunately, there is very little information on the composition of bile acids in Lp-X. According to S. Narayanan S., lithocholic acid is the main component of bile acids in Lp-X [47,48]. Unlike normal blood LPs that have a single PhL layer that surrounds the hydrophobic core of cholesterol esters and triglycerides, Lp-X has a vesicular structure [36]. 

Electron microscopic studies have revealed that Lp-X is a lamellar PhL bilayer or even multilayer particle with diameters of 30 to 70 nm, which has aggregate properties [36,49,50,51,52]. The Lp-X particle is characterized by a high level of PhLs (66 wt.%), mainly phosphatidylcholine and free, non-esterified cholesterol (22%), as well as by a low content of proteins (6%), cholesterol esters (3%), and triglycerides (3%) [53,54]. Albumin is a major protein component, which is located inside the core of Lp-X [54,55].

Lp-X also contains relatively small amounts of exchangeable apolipoproteins, such as apoA-1, apo-E, and apo-C, presumably bound to its PhL surface, but Lp-X does not contain apo-B. [36]. Apo-B is found in LDL and is a ligand for its receptors [44]. Due to the absence of Apo-B in the structure of Lp-X, this particle cannot interact with LDL receptors despite its similarity to LDL and does not, therefore, undergo hepatic clearance mediated by LDL receptors [38]. Moreover, there are reports on active Lp-X excretion from plasma by the kidneys [44,56]. The absence of Apo-B in the structure of Lp-X is most likely to be aimed at removing excess bile acids and cholesterol from the systemic circulation in PBC patients, bypassing the liver that is unable to fully perform its biliary function [44,56]. ApoA-1 is an activator of lecithin-cholesterol acyltransferase (LCAT). In PBC, the activity of this enzyme can be observed to be decreased [57,58,59]. A negligible quantity of esterified cholesterol in the Lp-X particle is induced by a small amount of ApoA-1 in its composition [41,42]. Plasma Lp-X concentration has been shown to be determined by the degree of cholestasis and by LCAT deficiency [44]. LCAT deficiency identified in patients with PBC requires the differential diagnosis of primary LCAT deficiency. In primary LCAT deficiency, a congenital defect, the presence of Lp-X is accompanied by a low concentration of HDL-C, anemia, corneal opacity, and impaired renal function [48].

The content of serum cholesterol in PBC is increased in the presence of Lp-X, which has a density similar to that of LDL. This makes Lp-X cholesterol indistinguishable from LDL-C by its quantification. Because of this similarity in density, Lp-X is often responsible for a false increase in LDL-C. Therefore, elevated LDL-C in patients with PBC requires careful interpretation and, if necessary, a test for Lp-X [36].

The mechanism and place of Lp-X formation in cholestasis have not been fully established [52]. It is speculated that bile is regurgitated into the plasma compartment as a consequence of cholestasis. As a result, on releasing into the plasma that does not contain bile acids and contains a higher albumin concentration than in the bile, LPs are rearranged to form Lp-X particles with a vesicular structure [36]. That is the reason why the ratio of albumin to bile acids in plasma is essential for the formation and maintenance of an Lp-X structural organization [37]. At the same time, in patients with PBC, the concentrations of PhLs and non-esterified cholesterol, which are determined in the LP complexes of the hepatic bile portion, are similar to those of PhL and free cholesterol in Lp-X [37]. However, at the same time, the amount of albumin increases in the newly formed Lps-X and there is attenuation of their bile acid concentration with a decrease in their amount to <0.01%, compared with their quantity (~1–3%) in biliary LPs [60]. There must be likely a redistribution of bile acids between biliary and serum LPs. Heimerl S. et al. report the identical content of bile acids up to <0.01% in LDL in patients with cholestasis due to the presence of Lp-X [60], while the same patients were seen to have no bile acids in HDL [60].

The hypothesis of Lp-X formation as a result of bile regurgitation into the blood is supported by the data obtained by Manzato et al., which have shown that biliary LP can be converted into “LP-X-like” material in vitro, by adding albumin or serum to native bile [37]. The Lp-X-like material formed in vitro has physicochemical and chemical characteristics similar or identical to the Lp-X isolated from serum. Conversely, in vitro incubation of Lp-X with bile acids can convert them into bile-LP-like particles [37]. In this connection, Lp-X is quoted as a combination of bile LP and albumin [61].

There may be also another mechanism of Lp-X formation. It seems likely that in PBC, bile acids can move into the systemic circulation not only with bile LPs, but their entry and presence in the blood will necessarily initiate the formation of complexes with PhLs and cholesterol due to the potent detergent properties of bile acids. The formation of these complexes requires additional synthesis of both PhLs and free (non-esterified) cholesterol. Fatty acids and orthophosphate are essential for the synthesis of PhLs. The data obtained by Heimerl S. et al. suggest that the hepatic synthesis of fatty acids and PhL is increased in cholestasis [60]. There are significantly increased PhL levels in cholestatic plasma compared to the control (6036 ± 1917 µM versus 1902 ± 492 µM) [60], as well as elevated levels of palmitic and oleic fatty acids [62]. These fatty acids are the main components of bile phosphatidylcholines (Figure 3).

PhLs and non-esterified cholesterol are contained in sufficient quantities in LDLs that are synthesized in the liver and are the main structures transporting cholesterol from the liver to peripheral tissues. This allows bile acids that have entered the systemic circulation due to cholestasis to initiate the solubilization of PhLs and cholesterol from LDL to form micellar and lamellar complexes and to involve plasma albumin in this process. This mechanism may be responsible for the increased hepatic synthesis of LDLs and their elevated plasma levels in PBC patients. The data obtained by Heimerl S. et al. suggest that the lipid composition of LDL and HDL in the plasma samples from cholestatic patients might be very similar to that of Lp-X, with a noticeable increase in the monounsaturated molecules of phosphatidylcholine (PC 32:1, PC 34:1), phosphatidylethanolamine (PE 32: 1, PE 34:1), and free cholesterol, with a simultaneous reduction in esterified cholesterol levels compared to the control [60]. These data indicate that Lp-X arises not only from bile regurgitation, but also from the higher supply of PhL and cholesterol, which are synthesized in the liver in response to the entry of bile acids into the systemic circulation [60]. The authors have shown that the higher hepatic synthesis of fatty acids and PhL plays an important role in the formation of Lp-X in cholestasis and that their entry into the systemic circulation most likely occurs in the vesicular form, and possibly together with free cholesterol [60]. That is probably why Lp-X does not inhibit hepatic de novo cholesterol synthesis in in vitro and in vivo, which has been noted in many studies [63,64,65].

## 5. Features of Lipid Metabolism Disorders in PBC

The plasma levels of palmitic and oleic acids, as well as PhLs and cholesterol, increase in patients with PBC just in its early stages [5,66]. Some authors have primarily attempted to consider the detectable increase in TC in PBC patients in terms of its effect on the development of atherosclerosis and pathological cardiovascular changes in these patients [32,43,67]. However, the elevated levels of cholesterol in patients with PBC, as well as those of phospholipids, are aimed at neutralizing the detergent effect of bile acids that have entered the systemic circulation as cholestasis progresses. In this regard, dyslipidemia in PBC is an “anomalous” condition and is the body’s compensatory response to the appearance of bile acids in systemic circulation. Therefore, despite the higher plasma TC levels in PBC patients, the latter is found in elevated HDL concentrations, poses a low risk for atherosclerosis and cardiovascular events, has a low-grade development of hepatic steatosis, and has the appearance of abnormal plasma Lp-X [25,31,67,68]. In patients suffering from primary biliary cholangitis, Lp-X is abundant; there is no increase in the incidence of cardiovascular events [43,44,69]. In this regard, PBC can serve as a model system for developing a new area in the search and design of drugs to treat dyslipidemia.

The studies conducted by Y. Zhang et al. show that patients with PBC have the lowest degree of hepatic steatosis not only among those with chronic liver diseases, but also have a lower degree than in healthy people [25]. The mechanism of low-grade concomitant hepatic steatosis is multifactorial. Insufficient intestinal bile acid release leading to fat malabsorption, steatorrhea, and gut microbial dysbiosis in PBC patients can cause a decrease in liver fat deposition [5,8]. In addition, the higher expression of FGF 19 induces a reduction in mitochondrial acetyl-coenzyme A-carboxylase 2, promoting free fatty acid oxidation and simultaneously inhibiting fatty acid synthesis, which lowers liver fat accumulation and plasma triglyceride levels [70].

The presence of Lp-X in liver diseases is of great clinical importance, since its detection is considered to be the most sensitive and specific biochemical marker of cholestasis [45]. A positive Lp-X test shows more than 95% agreement with histological methods used to confirm cholestatic syndrome [71]. However, due to the complexity of determining Lp-X and the high information value of biochemical markers, the determination of alkaline phosphatase, and γ-glutamyl transferase, a test for Lp-X is rarely used to diagnose the cholestatic state.

## 6. Conclusions

The developing disruption in the processes of bile secretion and enterohepatic circulation of bile salts in patients with PBC just in its stages leads to the insufficient release of bile acids into the bowel and to their entry into the systemic circulation, which is accompanied by dyslipidemia that is manifested by the elevated levels of TC, PhLs, fatty acids, LDL-C, and HLD-C. The mechanism of hyperlipidemia in cholestatic disorders, as well as in patients with PBC, differs from that in other conditions, since along with an elevation of TC, the HDL-C levels are increased and an unusual Lp-X appears. Lp-X, as well as other plasma LPs, should be considered as lipid transport structures in the systemic circulation, which emerge in response to the intake of bile acids in the blood plasma due to cholestasis. The formation of Lp-X is most likely the body’s protective reaction aimed at inactivating the detergent effect of bile acids on the membrane structures of blood corpuscles and vascular endothelial cells. It is bile acids, rather than TC levels, that must correlate with the level of Lp-X and determine its formation. Concomitant hypercholesterolemia in patients with PBC occurs without an increase in the incidence of atherosclerosis and cardiovascular events, since excess cholesterol in cholestatic conditions is aimed at neutralizing the detergent effect of bile acids that have entered the systemic circulation and is most probably to be a compensatory reaction of the body. “Anomalous” hypercholesterolemia in PBC can serve as a model system for the search and development of new methods for the treatment of dyslipidemia.

## Figures and Tables

**Figure 1 biomedicines-10-03046-f001:**
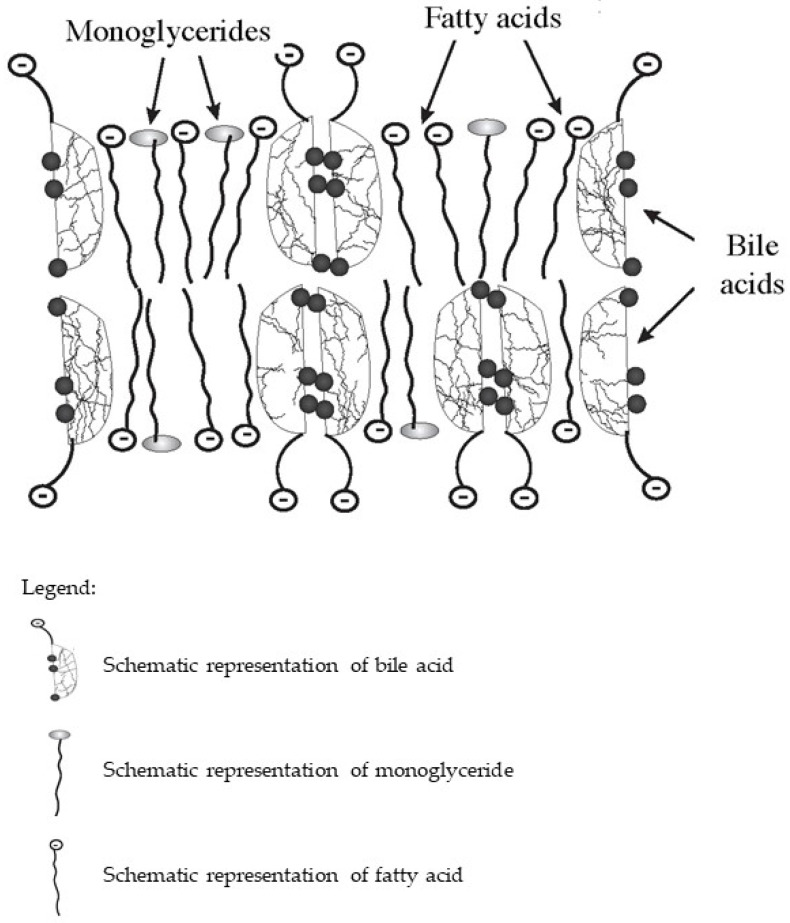
Schematic diagram of the composition of lipoid-bile complexes formed in the small bowel.

**Figure 2 biomedicines-10-03046-f002:**
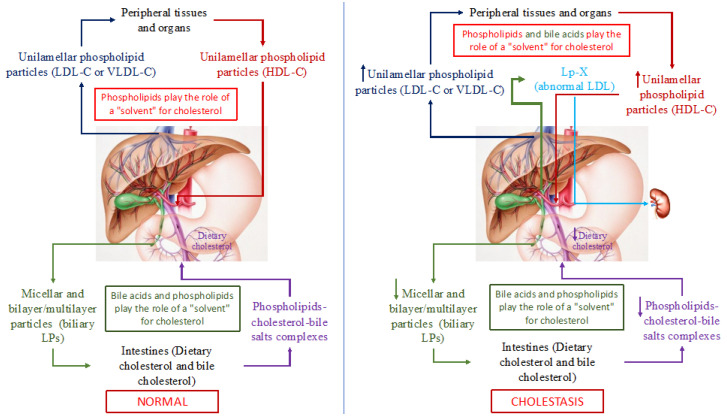
Schematic representation of cholesterol transport in the human body in normal (left part of the picture) and with cholestasis (right part of the picture). Note: LP(s)-lipoprotein(s); LDL-C—low-density lipoproteins cholesterol; VLDL-С—very low-density lipoproteins cholesterol; HDL-C—high-density lipoproteins cholesterol; Lp-X—lipoprotein X.

**Figure 3 biomedicines-10-03046-f003:**
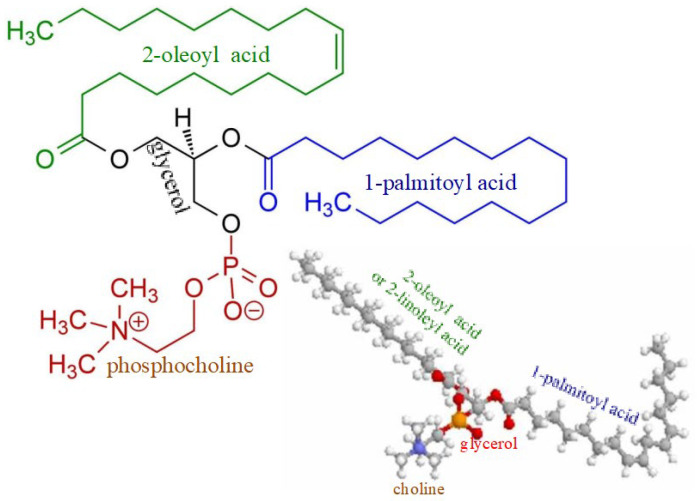
Chemical structure of phosphatidylcholine containing palmitic and oleic fatty acids.

## Data Availability

Not applicable.

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
