# Peer review of "Features of Lipid Metabolism Disorders in Primary Biliary Cholangitis"

_biomedicines, 2022, doi:10.3390/biomedicines10123046_

Round 1

Reviewer 1 Report

Dear Authors

Assessing one by one the article I would like to point out a few issues that need to be clarified/corrected.

The title contains an unauthorized division of the word cholangitis

Figure 1 is a bit inaccurate; it is not entirely clear which structures are indicated by the arrows. It should be corrected.

Is Fig 2 a legitimate borrowing of this picture? Or is it a photo taken by the authors? It should be recorded with the photo. Does the patient consent to this photo? If so, this issue should also be included under the photo so as not to raise doubts. If the patient is known to the authors - maybe a short description of the patient, with test results, would be helpful here.

The authors under figure 2 wrote that biochemical tests revealed ...? Does this apply to this patient? The authors use abbreviations for which I have not found explanations. Regardless, a table with the results would be useful here (even if not with values, for example, with arrows indicating the trends of changes) - it would improve the article's readability.

The section "The mechanism of dyslipidemia in PBC" could use a diagram. This text is difficult to read, and since it is a review article, charts and figures that facilitate understanding of the text are always welcome by readers because they allow you to organize your knowledge. I have no objections to the description of the mechanisms of dyslipidemia; they are presented correctly, are interesting, and testify to the researchers' insight.

In the "Features of lipid metabolism disorders in PBC" section, the authors bold the most important terms - but this is not recommended in the Journal, so I recommend removing this bold. Perhaps it is also worth considering a table or diagram (?)

To sum up, apart from one legal issue (Fig2) - described above and the technical aspects of the work, I have no substantive reservations.
The work is written correctly in terms of content but presented in a difficult way, which must undoubtedly be corrected.

Author Response

The authors express their sincere gratitude to the reviewer for the positive evaluation of our manuscript. All the comments of the reviewer have been taken into account and the following changes have been made to the manuscript:

  • The spelling of the word "cholangitis" has been corrected in the title
  • A note has been made for Figure 1 in which conditional images of bile acid, monoglyceride, fatty acid are indicated.
  • Figure 2 has been removed from the article, since obtaining written consent from the patient will currently take a long time, which will delay the consideration of the manuscript.
  • The figure "Schematic representation of cholesterol transport in the human body in normal (left part of the picture) and with cholestasis (right part of the picture)" has been added to the section "The mechanism of dyslipidemia in PBC"
  • All bold highlighting has been removed in the section "Features of lipid metabolism disorders in PBC"

The authors once again thank the reviewer for the comments aimed at improving the content of the manuscript

Reviewer 2 Report

Authors reviewed features of lipid metabolism disorders in primary biliary cholangitis.

1.      In Title, make change from “Chol-Angitis” to “Cholangitis”.

2.       In abstract section, “primary biliary cirrhosis is one of the autoimmune liver diseases.” PBC is relatively common diseases in Japan and US.

3.       Bezafibrate, which is one of the drugs for hyperlipidemia, is effective against PBC. Authors should mention about it. See: Kanda T, Yokosuka O, Imazeki F, Saisho H. Bezafibrate treatment: a new medical approach for PBC patients? J Gastroenterol. 2003;38(6):573-8. doi: 10.1007/s00535-002-1102-7.PMID: 12825134; Tanaka A. Current understanding of primary biliary cholangitis. Clin Mol Hepatol. 2021 Jan;27(1):1-21. doi: 10.3350/cmh.2020.0028. PMID: 33264835

Author Response

The authors express their sincere gratitude to the reviewer for the positive evaluation of our manuscript. All the comments of the reviewer have been taken into account and the following changes have been made to the manuscript:

Page 1, line 3, The spelling of the word "cholangitis" has been changed in the title

Page 1, line 7-8, “...a rare...” replaced by “...one of the...”

Page 8, line 431, links 68 and 69 have been added to the text;

Page 11, lines 629-632, added the links themselves

Round 2

Reviewer 1 Report

The article has been improved. Replies to my comments are entirely satisfactory.
Thanks to the authors for their efforts.

I believe the manuscript in its current version can be accepted and considered by the Editor for publication.